# Patterns and predictors of sick leave among Swedish non-hospitalized healthcare and residential care workers with Covid-19 during the early phase of the pandemic

**Marta A. Kisiel**[1]*, **Tobias Nordqvist**[1], **Gabriel Westman**[2], **Magnus Svartengren**[1], **Andrei Malinovschi**[3], **Helena Janols**[2]

1 Department of Medical Sciences, Occupational and Environment Medicine, Uppsala University, Uppsala, Sweden, 2 Department of Medical Sciences, Infectious Diseases, Uppsala University, Uppsala, Sweden, 3 Department of Medical Sciences, Clinical Physiology, Uppsala University, Uppsala, Sweden

* marta.kisiel@medsci.uu.se

**Data Availability Statement:** We uploaded the minimal anonymized data set necessary to

## Abstract

Healthcare and residential care workers represent two occupational groups that have, in particular, been at risk of Covid-19, its long-term consequences, and related sick leave. In this study, we investigated the predictors of prolonged sick leave among healthcare and residential workers due to non-hospitalized Covid-19 in the early period of the pandemic. This study is based on a patient register (n = 3209) and included non-hospitalized healthcare or residential care service workers with a positive RT-PCR for SARS-CoV-2 (n = 433) between March and August 2020. Data such as socio-demographics, clinical characteristics, and the length of sick leave because of Covid-19 and prior to the pandemic were extracted from the patient's electronic health records. Prolonged sick leave was defined as sick leave ≥ 3 weeks, based on the Swedish pandemic policy. A generalized linear model was used with a binary distribution, adjusted for age, gender, and comorbidity in order to predict prolonged sick leave. Of 433 (77% women) healthcare and residential care workers included in this study, 14.8% needed longer sick leave (> 3 weeks) due to Covid-19. Only 1.4% of the subjects were on sick leave because of long Covid. The risk of sick leave was increased two-fold among residential care workers (adjusted RR 2.14 [95% CI 1.31–3.51]). Depression/anxiety (adjusted RR 2.09 [95% CI 1.31–3.34]), obesity (adjusted RR 1.96 [95% CI 1.01–3.81]) and dyspnea at symptom onset (adjusted RR 2.47 [95% CI 1.55–3.92]), sick leave prior to the pandemic (3–12 weeks) (adjusted RR 2.23 [95% CI 1.21–4.10]) were associated with longer sick leave. From a public health perspective, considering occupational category, comorbidity, symptoms at onset, and sick leave prior to the pandemic as potential predictors of sick leave in healthcare may help prevent staff shortage.

replicate the study findings at Kaggle: https://www.kaggle.com/uppsala21/sick-leave-covid.

**Funding:** We have no external funders for this project. All the authors were employed at the Uppsala University Hospital and I have my research time provided by the Uppsala University Hospital.

**Competing interests:** The authors have declared that no competing interests exist.

## Introduction

Healthcare and residential care workers had the highest rate of SARS-CoV-2 positive cases compared to other occupational groups, particularly in the first wave of the pandemic [1–4]. A large population study from the U.K. found a seven-fold increased risk of Covid-19 among healthcare and two-fold higher risk among residential care workers [5]. Risk factors for Covid-19 among healthcare and residential care workers included inadequate personal protective equipment, suboptimal handwashing, and working long hours in high-risk departments [6–9].

Even though the majority of healthcare personnel had mild disease [1,10], up to 10% might have had symptoms that persisted beyond 12 weeks, called long Covid, reducing their work ability [11]. Studies from the U.S. and Spain showed that healthcare workers also had an increased prevalence of sick leave during the pandemic [12,13], a major driver of high healthcare costs [14]. Sick leave has a multifactorial association between morbidity, education, working conditions, job satisfaction, general health, and family life–as an indicator of individuals' well-being, but also a predictor of health consequences [15–17].

In Sweden, it was estimated that 3.4% of healthcare and 1.3% of residential care workers (per 100,000 people) had Covid-19 in the first wave of the pandemic [18]. Factors such as hospitalization, higher age, and sick leave prior to the pandemic were the risk factors of longer sick leave in the Swedish general population [5].

A recent editorial by Gohar suggested that staff shortage, increased work demand, and personal factors such as age, work experience, job role, health, history of previous sick leave, the organization safety, and employee job support were predictive factors for sick leave in healthcare [19]. In this study, we investigated predictors of prolonged sick leave after having a confirmed Covid-19 illness among non-hospitalized healthcare and residential care workers. The study covers the early period of the pandemic, when testing of both occupational categories was prioritized, providing valuable clinical and epidemiological information.

## Materials and methods

### Study population

This study is a part of COMBAT, a post Covid project investigating the long-term consequences of non-hospitalized Covid-19. It is is based on a patient register (n = 3209) containing information about symptomatic individuals who were tested at one Covid-19 testing outpatient center, created by the Department of Infectious Disease at the Uppsala University Hospital between March 10th and August 21st, 2020. This study population included symptomatic employees of Region Uppsala who tested positive for SARS-CoV-2 by reverse transcription–polymerase chain re-action (RT-PCR) on nasopharyngeal swab, not receiving in-patient care, and working in healthcare or residential care service. The included subjects were followed up after 8–12 months (until the end of April 2021). The Region Uppsala, being one of the regional governmental authorities responsible for public health and transportation, includes Uppsala University Hospital, primary healthcare, including residential care service, and primary care centers. Positive RT-PCR was the basis for a confirmed diagnosis of Covid-19 (U07.1), in accordance with the International Statistical Classification of Disease (ICD).

The study was approved by the institutional ethics committee of the University Hospital in Uppsala (2020–05707) and conducted in accordance with the Helsinki Declaration. All medical records were anonymized; only statistical information was used for research purposes.

### Framework: Swedish sick leave policy during the pandemic

The information for this study was further collected from the electronic medical records that contain all information on patients, readily available for healthcare providers in the Uppsala Region. The sick leave due to Covid-19 was divided into two groups: ≤ 3 weeks and > 3 weeks. Prolonged sickness absence was defined as sick leave > 3 weeks, based on the Swedish pandemic policy on sick leave. In line with this policy, all suspected and confirmed Covid-19 cases were advised to self-isolate, having a paid sick leave up to 21 days (≤ 3 weeks) from the symptom onset. From day 22 (> 3 weeks), a doctor's certificate was required for prolonged paid sick leave [20]. The group with sick leave > 3 weeks, defined as longer sick leave, included subjects who required a doctor's certificate. Some subjects had more than one doctor's certificate for their sick leave because of Covid-19, but there were no time gaps between them. Therefore, the length of sick leave in the group > 3 weeks was calculated as a period between a day of positive RT-PCR result and the last day of the doctor's certificate. We also assessed sick leave > 12 weeks.

### Potential predictors

Besides information on sick leave as a result of Covid-19, we collected the following data from the patient's medical records:

- Age (in years) and sex (woman/man), gathered from the Swedish national identification number.

- Information on occupation, gathered by a nurse at the time of RT-PCR testing and categorized as physician, nurse/midwives, assistant nurse, psychologist/counselor, biomedical analytics, occupational- and physiotherapist, residential care workers, health supporting staff such as medical administrators, laboratory workers, and maintenance workers.

- Symptom at onset, gathered by a nurse at the time of the RT-PCR testing, including fever, dyspnea, muscle and joint pain, impaired taste and smell, sore throat, headache, nasal symptoms, GI symptoms (including nausea, vomiting, diarrhea, stomach pain), fatigue resulting from Covid-19 illness, and pressure over chest/chest pain.

- History of common chronic disease (based on diagnostic code only from the last 10 years) including diabetes, hypertension, other heart disease, chronic lung disease including asthma and obstructive lung disease, cancer/immunosuppressive treatment, hypo/hyperthyreosis, depression/anxiety.

- BMI (kg/m2) measured by healthcare personnel in the last 5 years, classified as underweight < 18.5, normal 18.50 to 24.99, overweight between 25.0 and 29.99, and obesity > = 30.

- Sick leave prior to the pandemic, defined as sick leave, with a doctor's certificate from January 1, 2019 and February 28, 2020. In case of more than one sick leave period, the length was summed together and categorized as ≤ 3 weeks, 3–12 weeks, > 12 weeks of sick leave.

### Statistical analysis

The categorical variables were presented as number and frequency using percentages, and the continuous variables were presented as means with standard deviation (SD). Differences between the groups (≤ 3 weeks and > 3 weeks sick leave) were assessed using Chi2 test or Fisher's exact test (if the assumption for Chi2 test was not met). Multivariable generalized

linear models, with adjustment for selected factors such as age, sex and different comorbidities were used to examine the factors predicting longer sick leave (> 3 weeks) in comparison to sick leave ≤ 3 weeks. We selected these three confounding factors based on the literature and that they had no missing values. The results of the regression analysis were presented as relative risk (RR) with 95% confidence intervals (CI). We included the following independent variables in the model: age (as continues variable), sex (women vs man), BMI (obesity vs normal weight), different occupational groups (all subgroups as yes vs no), comorbidities (all subgroups as yes vs no), symptom at onset (all subgroups as yes vs no), sick leave one year before the pandemic (as three subgroups <3,3–12,>12 weeks). All variables were included in the multivariable model without any statistical variable selection. P < 0.05 was considered statistically significant. Covariates included in the model were chosen based on clinical and theoretical reasoning. All analysis was managed in Excel and SAS 9.4.

## Results

### Socio-demographic and clinical characteristics

This study included 433 non-hospitalized subjects with detectable SARS-CoV-2 by RT-PCR, representing healthcare and residential care workers, of which 335 (77%) were women. Sixty individuals (13.8%) had longer sick leave (> 3 weeks) due to confirmed Covid-19. All basic and clinical characteristics of the study population are shown in Table 1. The length of the complementary doctor's certificate was between 6–134 days (mean 43 days), S1 Fig. One subject was on sick leave at the end of the study period (241 days). Six subjects (1.4%) were on sick leave for > 12 weeks.

### Predictors of prolonged sick leave

Age (adjusted RR 1.00 [95% CI 0.98–1.02]) and sex (adjusted RR for women 1.08 [95% CI 1.00–1.17)] had no association with longer sick leave. Residential care workers had twice the risk of long sick leave because of Covid-19 (adjusted RR 2.14 [95% CI 1.31–3.51]), while the risk of sick leave was not increased in other healthcare categories. Among comorbidities, obesity (adjusted RR 1.96 [95% CI 1.01–3.81]) and depression/anxiety (adjusted RR 2.09 [95% CI 1.31–3.34]) were significantly associated with longer sick leave due to Covid-19. We did not consider the relative risk of being underweight since the number of samples in the group > 3 weeks was low. The presence of dyspnea at symptom onset (adjusted RR 2.47 [95% CI 1.55–3.92]) was the only symptom that predicted longer sick leave. Sick leave prior to the pandemic (3–12 weeks) led to an increased risk of sick absence as a result of Covid-19, Fig 1.

## Discussion

To the best of our knowledge, this is the first study investigating the pattern and predictors of sick leave among healthcare and residential care workers with non-hospitalized confirmed Covid-19 in the early period of the pandemic. Our main result was that working at residential care exhibited almost two-fold higher risk for longer sick leave (< 3 weeks), while other healthcare groups had no elevated risk of sick absence due to Covid-19. Our finding is in line with research conducted prior to the pandemic, showing that residential care workers had the highest number of cases with long-term sickness absence compared to other professional categories [21]. In those studies, identified risk factors included: having a stressful work environment, poor work support, shortage of staff, increased mental and physical fatigue, and low job control [19,22,23]. In Sweden, the media highlighted that at the beginning of the pandemic, the

**Table 1. Socio-demographic and clinical characteristics of the healthcare and home/service care workers divided into two groups as: Sick leave ≤3 weeks and sick leave >3 weeks.**

|  | ≤ 3 weeks n = 373 | >3 weeks n = 60 | P |
|---|---|---|---|
| **Sex (women)** | 283 (84.5) | 52 (15.5) | 0.06 |
| **Age, mean (SD)** | 40.44 (13.30) | 42.34 (12.08) | 0.07 |
| **Occupational group** |  |  |  |
| Physician | 42 (82.3) | 9 (17.7) | 0.40 |
| Nurse/Midwife | 114 (90.5) | 12 (9.5) | 0.12 |
| Assistant nurse | 87 (89.7) | 10 (10.3) | 0.25 |
| Physio- and occupational therapist | 8 (80.0) | 2 (20.0) | 0.63 |
| Psychologist/ curator | 6 (85.7) | 1 (14.3) | 0.99 |
| Biomedical analytics | 6 (85.7) | 1 (14.3) | 0.99 |
| No patient contact | 54 (87.8) | 6 (12.2) | 0.42 |
| Residential care | 53 (74.6) | 18 (25.4) | **<0.01** |
| **Comorbidity** |  |  |  |
| Hypertension | 34 (87.2) | 5 (12.8) | 0.71 |
| Diabetes | 8 (80.0) | 2 (20.0) | 0.63 |
| Hypo-/hyperthyroidism | 29 (90.6) | 3 (0.4) | 0.59 |
| Heart disease | 18 (81.8) | 4 (18.2) | 0.52 |
| Lung disease | 35 (85.3) | 6 (14.7) | 0.87 |
| Cancer/Immunosuppressive treatment | 15 (100) | 0 | 0 |
| Depression/anxiety | 87 (76.9) | 26 (23.1) | **<0.01** |
| **Comorbidity** |  |  |  |
| No disease | 208(88.5) | 27(11.5) | 0.12 |
| One disease | 111(83.5) | 22(16.5) | 0.27 |
| Two or more diseases | 54(83.1) | 11(16.9) | 0.42 |
| **BMI** |  |  |  |
| Underweight <18.5 | 3 (60.0) | 2 (40.0) | 0.17 |
| Normal weight 18.5–25 | 102 (87.2) | 15 (12.8) | 0.26 |
| Overweight >25–30 | 62 (87.3) | 9 (12.7) | 0.44 |
| Obesity >30 | 25 (73.5) | 9 (26.5) | 0.05 |
| **Symptom at onset** |  |  |  |
| Fever | 174 (84.5) | 32 (15.5) | 0.33 |
| Dyspnea | 65 (73.8) | 23 (26.2) | **<0.01** |
| Muscle and joint pain | 173 (83.2) | 35 (16.8) | 0.85 |
| Impaired taste and smell | 93 (94.9) | 5 (5.1) | **<0.01** |
| Sore throat | 110 (83.3) | 22 (16.7) | 0.29 |
| Headache | 157 (84.8) | 28 (15.2) | 0.50 |
| Nasal symptoms | 253 (85.2) | 44 (14.8) | 0.39 |
| GI symptoms | 23 (82.1) | 5 (17.9) | 0.56 |
| Fatigue | 90 (91.8) | 8 (8.2) | 0.06 |
| Pressure over chest | 24 (80.0) | 6 (20.0) | 0.31 |
| **The length of sick leave prior to the pandemic (January 2019–February 2020)** |  |  |  |
| <3 week | 39 (70.9) | 16 (29.1) | 0.49 |
| 3–12 weeks | 27 (73.0) | 10 (27.0) | 0.24 |
| >12 weeks | 9 (75.0) | 3 (25.0) | 0.58 |

The P- values <0.05 of significance. BMI–missing data for 206 subjects. Occupation groups–missing data for 4 subjects.

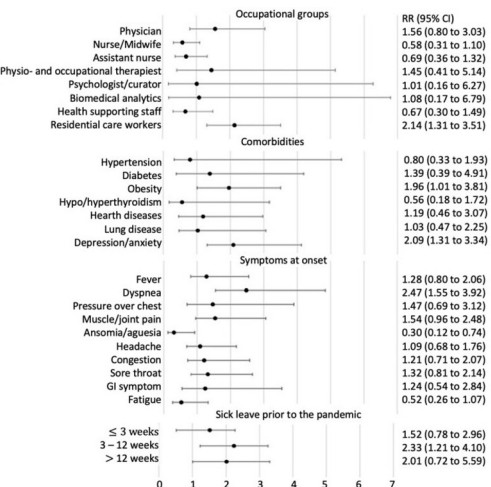

**Fig 1. The predictors of longer sick leave >3 weeks was determined by relative risk (RR) with confidence interval (95% CI).** The RR values were adjusted for age, gender, and comorbidity.

work demand was increased, and there was limited availability of protective equipment in residential care service [24,25].

In this study, we found that the relative risk for longer sick leave was not statistically different between the genders; however, women were more likely to take longer sick leave in comparison to men. This is in line with the previous research showing that women received more sickness benefits due to Covid-19 than men [26]. Also, in accordance with the Swedish report prior to the pandemic, more females, in general, took sick leave than males [27]. It was suggested that women more likely suffered long-term sequel after Covid-19, whereas men were at a higher risk for severe Covid-19 and its complications [28]. In addition, our study population mainly comprised women, especially among working categories of nurses, assistant nurses, and residential care personnel. These occupational groups have also been the most exposed while working closely with Covid-19 infected patients [1].

We found that depression and anxiety were predictors of prolonged sick leave. A study from China showed that approximately one-fifth of hospitalized Covid-19 patients had depression and anxiety [29]. A study conducted prior to the pandemic found that mental health disease was an important factor in longer and repeated sick leave, particularly in younger men and women [30,31].

As illustrated in our generalized linear model, obesity (> 30) was linked to a greater risk of longer sick absence. This finding is congruent with several previous studies showing that obese individuals have a higher likelihood of sickness absence compared to normal weight individuals [32,33]. Obesity was also associated with increased risk of hospitalization due to Covid-19, and critical outcomes from the infection, particularly in younger people [34–36].

Having dyspnea as a baseline symptom was a factor that predicted longer sick leave in our study. This is in line with a previous study where dyspnea at baseline was the only predictor affecting the recovery, both in outpatients and admitted patients [37]. The meta-analysis on Covid-19 showed that dyspnea as a symptom onset was strongly associated with progression of Covid-19 and risk of hospitalization [38].

Further, our study found that 3–12 weeks of sick leave prior to the pandemic increased the risk of prolonged sick absence due to a Covid-19 infection. This effect was not found in those with > 12 weeks of sick leave prior to the pandemic, which might be because of the low sample of subjects. A recent study on sick leave as a result of Covid-19 in the general population

reported that sick leave prior to the pandemic was linked to a longer sick leave due to Covid-19 [26]. Also, studies conducted prior to the pandemic showed that healthcare workers who had been on previous sick leave were also at risk of future absenteeism [23,39,40].

In this study, we found that sick leave > 12 weeks was uncommon and affected less than 2% of the studied non-hospitalized healthcare and residential care workers. In contrast, a study on the Swedish general population, based on register data of both hospitalized and non-hospitalized patients, showed that long-term lingering symptoms following Covid-19 affected 13% of individuals [26]. Also, the previous study on Swedish healthcare showed that eight months after mild Covid-19, one in ten individuals still had at least one moderate to severe symptoms, so called long Covid, which negatively impacted their daily life and work ability [11].

The study's strength is that it is based on all healthcare and residential care workers with confirmed Covid-19, tested during the first wave of the pandemic in the same testing center in Uppsala Sweden. Clinical characteristics were collected from the medical records, which reduces bias in comparison to self-reported information. By using patients' medical records, instead of the Swedish Insurance System, sick leave information was assessed in real time (until the end of April 2021) without delay. However, collecting sick leave information in this way might bias the study, as Swedish inhabitants are free to seek healthcare in different regions. The various regions in Sweden have distinct patient records. Another limitation is that we have a limited number of predictors. In addition, BMI information was available in the electronic medical records for only 53% of the subjects. A previous study showed that health professionals mostly recorded information about BMI, for patients with deviating weight and if it was clinically relevant [41], which might also bias our study. Furthermore, the statistical power to examine some predictors, including occupational groups, was low. Therefore, we could only control for a limited number of confounders, such as age, gender, and comorbidity. It has previously been found that these factors significantly influence the occurrence of sickness absence [22].

## Conclusions

In this study, we showed that working in residential care, obesity, depression/anxiety, and longer sick leave prior to the pandemic were predictors of sick leave in confirmed, non-hospitalized, Covid-19 cases. The pandemic is not yet over; from a public health perspective, identification of predictors of sick leave due to Covid-19 can be a coping strategy, preventing long sick leave or as an indicator of more severe disease. Thus, if front-line workers are affected, it may lead to staff shortage. Further studies with more sick leave predictors, comparing both hospitalized and non-hospitalized Covid-19 patients, are needed.

## Supporting information

**S1 Fig. The length (days) of sick leave due to Covid-19 with doctor's certificate in the group > 3 weeks.**
(PDF)

## Acknowledgments

The authors would like to acknowledge medical students Tove Wikström and Hanna Broman for helping with data collection.

## Author Contributions

**Conceptualization:** Marta A. Kisiel, Magnus Svartengren, Helena Janols.

**Data curation:** Marta A. Kisiel, Helena Janols.

**Formal analysis:** Marta A. Kisiel, Tobias Nordqvist.

**Investigation:** Marta A. Kisiel.

**Methodology:** Marta A. Kisiel, Tobias Nordqvist, Gabriel Westman, Magnus Svartengren, Andrei Malinovschi, Helena Janols.

**Project administration:** Marta A. Kisiel.

**Resources:** Andrei Malinovschi.

**Writing – original draft:** Marta A. Kisiel, Andrei Malinovschi, Helena Janols.

**Writing – review & editing:** Marta A. Kisiel, Tobias Nordqvist, Gabriel Westman, Magnus Svartengren, Helena Janols.

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
