## [Decision Letter · Decision Letter 0]

2 Aug 2021

PONE-D-21-21612

Patterns and predictors of sick leave among non-hospitalized healthcare and residential care workers with Covid-19 during the early phase of the pandemic

PLOS ONE

Dear Dr. Kisiel,

Thank you for submitting your manuscript to PLOS ONE. After careful consideration, we feel that it has merit but does not fully meet PLOS ONE’s publication criteria as it currently stands. Therefore, we invite you to submit a revised version of the manuscript that addresses the points raised during the review process.

We look forward to receiving your revised manuscript.

Kind regards,

Huei-Kai Huang, M.D.

Academic Editor

PLOS ONE

Journal Requirements:

5. Please include your table as part of your main manuscript and remove the individual files. Please note that supplementary tables (should remain/ be uploaded) as separate "supporting information" files

Additional Editor Comments:

The methodological issues raised by the reviewers should be appropriately addressed in your revised manuscript. Please note that additional reviewer comments could be found in the attached document.

Reviewers' comments:

Reviewer's Responses to Questions

**Comments to the Author**

1. Is the manuscript technically sound, and do the data support the conclusions?

Reviewer #1: Yes

Reviewer #2: Partly

Reviewer #3: Yes

2. Has the statistical analysis been performed appropriately and rigorously? 

Reviewer #1: Yes

Reviewer #2: I Don't Know

Reviewer #3: Yes

3. Have the authors made all data underlying the findings in their manuscript fully available?

Reviewer #1: Yes

Reviewer #2: Yes

Reviewer #3: No

4. Is the manuscript presented in an intelligible fashion and written in standard English?

Reviewer #1: Yes

Reviewer #2: No

Reviewer #3: Yes

5. Review Comments to the Author

**Reviewer #1:** Thank you for allowing me to review your study, “Patterns and predictors of sick leave among non-hospitalized healthcare and residential care workers with Covid-19 during the early phase of the pandemic”. The study examined predictors of sick leave among non-hospitalized healthcare & residential care workers from March to August 2020.

Overall, my impressions of this paper are positive. It offers meaningful data on a population exposed to various health risks and, naturally, higher than average sickness absence rates. I have included some remarks to help ensure the success of this paper.

• To improve this paper, I believe that more editing is required. I list some examples below.

++ “The study from the US and Spain showed that healthcare workers also had increased prevalence of sick leave during the pandemic….” It is unclear by stating “the study” that it was one study or several studies.

++ Issues with the structure of the sentence beginning with “A recent editorial by Gohar…”, specifically with “job support employees.” Do you mean “employee job support?”

++ Inconsistencies “healthcare” and “health care.”

++ “However, a limitation is that there might be a bias by collecting sick leave information in this way as Swedish inhabitants are free to seek health care in different regions that have distinct patients records BMI was missing in the substantial part of the study population.” I am unclear if these are supposed to be two separate sentences. If not, please clarify your point.

• It is indicated that multiple sick leave periods were summed together and later categorized based on total duration. Were there any measures taken for sick leaves that were uncommon (e.g., car accident)?

• How updated are medical records concerning variables such as BMI? If not updated regularly, I believe this should be described in the limitation section. Also, is BMI self-reported in these records? If it is self-reported, it is an important consideration to add as individuals typically underestimate their weight or BMI (which further magnifies the role of BMI and sick leave).

• It was mentioned that BMI was missing in a substantial part of the study population. Could this have compromised the integrity of the results?

• Age in previous studies showed inconsistencies with its relationship to sickness absence. However, some evidence suggests that sex (females) have higher rates of sick leaves. It might be worthy of hypothesizing why there was not a relationship. Could it be that COVID-19 affects male and female healthcare workers evenly? I would draw on the statistics (sex) of patients infected with COVID in the Uppsala region and contrast accordingly.

• Also missing from this study is its implications. By having this information, what should be done in terms of future research or from a practical standpoint?

• Mentioning the region or at least the country in the title is beneficial

My best wishes!

**Reviewer #2: **This is an interesting, important and generally well structured study. However, there is a fair amount of copy editing and revision for clarity of language that needs to be done before it can be published, in my opinion. I have attempted to make some suggestions along these lines in the attached document. My grammatical and wording edits are only suggestions, but I think they will add to the readability of the paper.

I also had some concerns about the description of the statistical methods and presentation of the results which I have also detailed in the attached document. These, I think, must be addressed.

Thanks you for sharing your work; I enjoyed reading it.

**Reviewer #3:** This is an interesting study. The manuscript is well-written. However, some methodologic issues should be further clarified. My specific comments are as follows:

1. The authors only mentioned that BMI data are missing for 206 subjects. How about the missing data regarding other variables (listed in Table 1) used in the prediction model? Please also clarify this in the manuscript.

2. The definitions of the variables (predictors) or how they were identified were not clearly described in the Method section. For instance, how the symptoms at onset were defined or identified (e.g., what GI symptoms? How to define fatigue?)? How were the comorbidities defined (e.g., using diagnostic codes only? any criteria for diagnosis times or the time period to identify the comorbidities?)? This section should be revised to provide more details.

3. The study sample size is only 433 (≤ 3 weeks, n=373; >3 weeks, n=60). The patient no. of many predictors is low; some were < 10 (as shown in Table 1). Whether the statistical power was enough to evaluate the effect of those predictors should be doubted.

4. According to the Statistical analysis section, the regression models only adjusted for age, gender, and comorbidities, but not other variables in Table 1, based on clinical and theoretical reasoning. However, to me, the reasons for choosing only age, gender, and comorbidities for adjustment seem arbitrary. Is it possible to perform additional analyses (sensitivity analyses) that applied other variable selection methods (e.g., stepwise selection method, LASSO regression)?

6. PLOS authors have the option to publish the peer review history of their article (what does this mean?). If published, this will include your full peer review and any attached files.

Reviewer #1: No

Reviewer #2: **Yes: **Matthew Groenewold

Reviewer #3: No

---

## [Author Response · Author response to Decision Letter 0]

13 Oct 2021

Dear Academic Editor Dr. Huang,

Thank you for the constructive comments! 

Below please find our responses to your comments and changes made in the manuscript as a result. 

Reviewer #1: Thank you for allowing me to review your study, “Patterns and predictors of sick leave among non-hospitalized healthcare and residential care workers with Covid-19 during the early phase of the pandemic”. The study examined predictors of sick leave among non-hospitalized healthcare & residential care workers from March to August 2020.

Overall, my impressions of this paper are positive. It offers meaningful data on a population exposed to various health risks and, naturally, higher than average sickness absence rates. I have included some remarks to help ensure the success of this paper.

• To improve this paper, I believe that more editing is required. I list some examples below.

Answer: Thank you for the comment. An English proofreader has now reviewed the manuscript, and we have taken into account all the comments below as well.

++ “The study from the US and Spain showed that healthcare workers also had increased prevalence of sick leave during the pandemic….” It is unclear by stating “the study” that it was one study or several studies.

Answer: Thank you for the comment. We have now corrected this in the introduction; see the second paragraph. 

++ Issues with the structure of the sentence beginning with “A recent editorial by Gohar…”, specifically with “job support employees.” Do you mean “employee job support?”

Answer: Thank you for this important comment. We have revised the introduction; see the fourth paragraph. 

++ Inconsistencies “healthcare” and “health care.”

Answer: We apologize for this inconsistency. We have now used ‘healthcare’ consistently throughout the manuscript. 

++ “However, a limitation is that there might be a bias by collecting sick leave information in this way as Swedish inhabitants are free to seek health care in different regions that have distinct patients records BMI was missing in the substantial part of the study population.” I am unclear if these are supposed to be two separate sentences. If not, please clarify your point.

Answer: We apologize for this inconsistency. These should be stated as two sentences (the last paragraph of the discussion section), which we have now corrected in the manuscript. 

• It is indicated that multiple sick leave periods were summed together and later categorized based on total duration. Were there any measures taken for sick leaves that were uncommon (e.g., car accident)?

Answer: Thank you for this interesting comment. We understand that you are asking about sick leave prior to the pandemic. There were no uncommon reasons for sick leave; therefore, we find this reason as being unlikely to introduce a bias.

• How updated are medical records concerning variables such as BMI? If not updated regularly, I believe this should be described in the limitation section. Also, is BMI self-reported in these records? If it is self-reported, it is an important consideration to add as individuals typically underestimate their weight or BMI (which further magnifies the role of BMI and sick leave).

Answer: Thank you for this comment. Information on BMI was collected from the electronic medical records of patients. Height and weight used to be measured therefore by healthcare personnel during in- and outpatient visits. Therefore, there is a bias with regard to collection of BMI as this was related to healthcare contact. This is discussed now in Material and Methods (page 7) and the limitation section (page 12). 

• It was mentioned that BMI was missing in a substantial part of the study population. Could this have compromised the integrity of the results?

Answer: This is an important comment. Please see the answer above. 

• Age in previous studies showed inconsistencies with its relationship to sickness absence. However, some evidence suggests that sex (females) have higher rates of sick leaves. It might be worthy of hypothesizing why there was not a relationship. Could it be that COVID-19 affects male and female healthcare workers evenly? 

Answer: Thank you for the comment. We have added a paragraph in the discussion, page 10. 

I would draw on the statistics (sex) of patients infected with COVID in the Uppsala region and contrast accordingly.

Answer: Thank you for this interesting comment. We agree that this comparison would be relevant. However, this estimate in the population is unreliable because the accessibility for Covid-19 testing was low in the general population in the first part of the pandemic. The study covers the early period of the pandemic, where testing of both occupational categories was prioritized, providing valuable clinical and epidemiological information (see the introduction, page 5).

• Also missing from this study is its implications. By having this information, what should be done in terms of future research or from a practical standpoint?

Answer: Thank you for the comment. We have added implications of the study at the end of the last paragraph of the discussion, page 12, as well as in the conclusion, page 12.

• Mentioning the region or at least the country in the title is beneficial

Answer: Thank you for the comment. We have modified the title as you suggested. 

My best wishes!

Reviewer #2: This is an interesting, important and generally well-structured study. However, there is a fair amount of copy editing and revision for clarity of language that needs to be done before it can be published, in my opinion. I have attempted to make some suggestions along these lines in the attached document. My grammatical and wording edits are only suggestions, but I think they will add to the readability of the paper.

I also had some concerns about the description of the statistical methods and presentation of the results which I have also detailed in the attached document. These, I think, must be addressed.

Thanks you for sharing your work; I enjoyed reading it.

Answer: Thank you for your suggestions for the manuscript. We have made changes according to your comments; please look at the manuscript: 

Reviewer #3: This is an interesting study. The manuscript is well-written. However, some methodologic issues should be further clarified. My specific comments are as follows:

1. The authors only mentioned that BMI data are missing for 206 subjects. How about the missing data regarding other variables (listed in Table 1) used in the prediction model? Please also clarify this in the manuscript.

Answer: Thank you for this important comment. In the prediction model, we used age, gender, and comorbidity where we did not have missing data. Age and gender were determined from the Swedish national identification number, while comorbidity was gathered from the electronic medical records. However, underdiagnosis or that some diagnoses were not registered in the medical journals cannot be excluded. 

2. The definitions of the variables (predictors) or how they were identified were not clearly described in the Method section. For instance, how the symptoms at onset were defined or identified (e.g., what GI symptoms? How to define fatigue?)? How were the comorbidities defined (e.g., using diagnostic codes only? any criteria for diagnosis times or the time period to identify the comorbidities?)? This section should be revised to provide more details.

Answer: Thank you for the comment. We have clarified this in the materials and methods, pages 6-7. 

3. The study sample size is only 433 (≤ 3 weeks, n=373; >3 weeks, n=60). The patient no. of many predictors is low; some were < 10 (as shown in Table 1). Whether the statistical power was enough to evaluate the effect of those predictors should be doubted.

Answer: Thank you for this comment. Our study has several limitations, including the low number of some variables/predictors. Please see the discussion and the limitations, page 12.

4. According to the Statistical analysis section, the regression models only adjusted for age, gender, and comorbidities, but not other variables in Table 1, based on clinical and theoretical reasoning. However, to me, the reasons for choosing only age, gender, and comorbidities for adjustment seem arbitrary. Is it possible to perform additional analyses (sensitivity analyses) that applied other variable selection methods (e.g., stepwise selection method, LASSO regression)?

Answer: Thank you for this comment. We adjusted for age, gender, and comorbidities because they are the factors that were previously shown to influence sick leave (Allebeck et al., 2004). We did not adjust for factors that were missing in the substantial part of subjects (as BMI) or had a small sample size (as some occupational groups). Please see the discussion section and the limitations, page 12. 

Based on the literature on mythos and misinterpretation in statics, we chose not to use the stepwise selection method or LASSO regression as these methods may complicate the analysis, in cases of variables with small sample size by invalidating common tools of statistical inference such as P-values and confidence intervals (Heize et al., Five myths about variable selection Transplant international, 30(1) 2016; Greenland et al., Statystical tests, P value, confidence intervals, and power: a guide to misinterpretation. Eur J Epidemiol, 31:337-350, 2016). We emphasized that our results should be interpreted with a caution, and we made bullet points for all the limitations in the last paragraph of the discussion section. 

6. PLOS authors have the option to publish the peer review history of their article (what does this mean?). If published, this will include your full peer review and any attached files.

Do you want your identity to be public for this peer review? For information about this choice, including consent withdrawal, please see our Privacy Policy.

Reviewer #1: No

Reviewer #2: Yes: Matthew Groenewold

Reviewer #3: No

 Accepting review changes in the manuscript:

---

## [Decision Letter · Decision Letter 1]

15 Nov 2021

Patterns and predictors of sick leave among Swedish non-hospitalized healthcare and residential care workers with Covid-19 during the early phase of the pandemic

PONE-D-21-21612R1

Dear Dr. Kisiel,

We’re pleased to inform you that your manuscript has been judged scientifically suitable for publication and will be formally accepted for publication once it meets all outstanding technical requirements.

Kind regards,

Huei-Kai Huang, M.D.

Academic Editor

PLOS ONE

Additional Editor Comments (optional):

Reviewers' comments:

Reviewer's Responses to Questions

**Comments to the Author**

1. If the authors have adequately addressed your comments raised in a previous round of review and you feel that this manuscript is now acceptable for publication, you may indicate that here to bypass the “Comments to the Author” section, enter your conflict of interest statement in the “Confidential to Editor” section, and submit your "Accept" recommendation.

Reviewer #1: All comments have been addressed

Reviewer #2: All comments have been addressed

Reviewer #3: All comments have been addressed

2. Is the manuscript technically sound, and do the data support the conclusions?

Reviewer #1: Yes

Reviewer #2: Yes

Reviewer #3: Yes

3. Has the statistical analysis been performed appropriately and rigorously? 

Reviewer #1: I Don't Know

Reviewer #2: Yes

Reviewer #3: Yes

4. Have the authors made all data underlying the findings in their manuscript fully available?

Reviewer #1: Yes

Reviewer #2: Yes

Reviewer #3: Yes

5. Is the manuscript presented in an intelligible fashion and written in standard English?

Reviewer #1: Yes

Reviewer #2: Yes

Reviewer #3: Yes

6. Review Comments to the Author

Reviewer #1: The authors have adequately resolved all my comments in their resubmission. I believe that the paper is worthy of publication.

Reviewer #2: (No Response)

Reviewer #3: (No Response)

7. PLOS authors have the option to publish the peer review history of their article (what does this mean?). If published, this will include your full peer review and any attached files.

Reviewer #1: No

Reviewer #2: No

Reviewer #3: No

---

## [Editor Report · Acceptance letter]

1 Dec 2021

PONE-D-21-21612R1 

Patterns and predictors of sick leave among Swedish non-hospitalized healthcare and residential care workers with Covid-19 during the early phase of the pandemic 

Dear Dr. Kisiel:

I'm pleased to inform you that your manuscript has been deemed suitable for publication in PLOS ONE. Congratulations! Your manuscript is now with our production department. 

Kind regards, 

on behalf of

Dr. Huei-Kai Huang 

Academic Editor

PLOS ONE